# ExploraQA: Embodied Question Answering with Long-horizon Proactive Exploration

## Abstract

Embodied Question Answering (EQA) is a critical task for developing embodied intelligence, requiring agents to autonomously explore environments and answer human questions through perception, navigation, and reasoning. However, existing EQA benchmarks suffer from three key limitations: constrained exploration scope, passive trajectory, and insufficient viewpoint annotation. To address these challenges, we introduce ExploraQA, a large-scale dataset featuring 12,436 diverse, open-ended questions across seven categories, designed to evaluate language, visual, and spatial reasoning. ExploraQA emphasizes long-horizon exploration, proactive trajectory, and comprehensive viewpoint annotations, enabling rigorous assessment of autonomous agents. We further propose an Iterative EQA Data Generation Framework to efficiently produce high-quality annotations via VLMs and human verification. To enhance exploration, we present the Answer Quality-Guided Navigator, which leverages a Topology-Aware Keyframe Search Module for efficient long-range navigation and an Answer Quality Reward Mechanism to optimize question-driven trajectories through dual LLM evaluators. Experimental results show that AQ-Nav achieves a 5.4% absolute improvement in $E\_score$ on the ExploraQA unseen test set over state-of-the-art navigators. We will release our dataset and code.

## 1 Introduction

Embodied Question Answering (EQA) has emerged as a crucial task within the broader pursuit of embodied intelligence. Concretely, the EQA task requires agents to autonomously explore and analyze their environment to answer human questions, involving perception, navigation, and reasoning capabilities.

Current EQA benchmarks Das et al. (2018a); Yu et al. (2019); Majumdar et al. (2024); Wijmans et al. (2019); Gordon et al. (2017) exhibit three major limitations: *(1) Constrained Exploration Scope:* Existing datasets predominantly focus on short-range exploration, with average distances of merely 1-2m as shown in Tab. 1, while neglecting the challenges of long-range exploration. Such constrained settings fail to reflect scenarios where agents must navigate expansive environments to locate distant targets to answer questions; *(2) Passive Trajectory:* Some benchmarks Majumdar et al. (2024) employ a two-stage process: first generating exploration trajectories through question-agnostic methods (e.g., Frontier-Based Exploration in Fig. 1), second constructing QA pairs based on the visual observations captured along these trajectories. This decoupled approach produces suboptimal trajectories that are not tailored to specific questions, rendering them ineffective as expert demonstrations for imitation learning; *(3) Insufficient Viewpoint Annotation:* Existing benchmarks Das et al. (2018a;b) annotate only a single viewpoint as the ground-truth answer location. However, trajectories often contain multiple valid viewpoints that could answer the same question. This sparse annotation promotes overfitting to endpoint biases and constrains valid alternatives, limiting proactive reasoning.

To address these limitations, we present **ExploraQA**, a novel, large-scale dataset featuring three distinctive characteristics: *(1) Long-horizon Exploration Trajectories:* The benchmark requires agents to perform extensive navigation, with an average trajectory length of 10.4m, a more than five-fold increase over prior work (1.9m). Such long-horizon design necessitates efficient environment exploration and effective information gathering; *(2) Proactive Trajectory:* ExploraQA integrates

Table 1: **Comparison of the EQA datasets.** 'Open Vocab' indicates that the dataset's questions and answers are not restricted to a predefined vocabulary. 'Active' indicates that the agent can interact with the environment. 'LLM score' signifies that a Large Language Model is used to evaluate the quality of the answers.

| Datasets | Capabilities | | | | Statistics | |
| --- | --- | --- | --- | --- | --- | --- |
| | Open Vocab | Active | LLM score | Proactive Traj. | Traj. Len. | Total Ques. |
| SQA3D [ICLR 2022] | ✓ | ✗ | ✗ | ✗ | - | 33.4k |
| RoboVQA [NIPS 2023] | ✓ | ✗ | ✗ | ✗ | - | 829k |
| MMbench [ECCV 2024] | ✓ | ✗ | ✓ | ✗ | - | 2.9k |
| IQA [CVPR 2017] | ✗ | ✓ | ✗ | ✗ | - | 75k |
| EQA-v1 [CVPR 2018] | ✗ | ✓ | ✗ | ✗ | 1.9m | 5.2k |
| MP3D-EQA [CVPR 2019] | ✗ | ✓ | ✗ | ✗ | 1.9m | 1.1k |
| MT-EQA [CVPR 2019] | ✗ | ✓ | ✗ | ✗ | 1.9m | 19k |
| OpenEQA [CVPR 2023] | ✓ | ✓ | ✓ | ✗ | - | 0.5k |
| ExploraQA (Ours) | ✓ | ✓ | ✓ | ✓ | 10.4m | 12k |

expert-annotated trajectories with corresponding language instructions from Krantz et al. (2020) to construct QA pairs as shown in Fig. 2. This ensures that agents can dynamically explore environments in a question-guided manner, efficiently reaching relevant regions to answer the question; *(3) Comprehensive Viewpoint Annotation:* The benchmark annotated multiple valid viewpoints for each question. This dense annotation method encourages agents to learn diverse observational strategies, rewarding agents for reaching any valid viewpoint rather than enforcing convergence to a single predefined location.

Additionally, ExploraQA includes 12,436 diverse, open-ended questions spanning seven distinct categories: object recognition, state recognition, style recognition, functional reasoning, commonsense reasoning, spatial localization and spatial reasoning. This dataset serves as a comprehensive benchmark for embodied research, fostering advancements in autonomous agent performance.

To facilitate large-scale EQA data annotation, we introduce **Iterative EQA Data Generation Framework**. This approach leverages VLMs in a cyclical process of generation and validation, progressively refining question-answer pairs across multiple iterations. The methodology enables efficient and scalable EQA dataset construction. Additionally, all generated data undergoes human verification to ensure high-quality annotations.

To enhance the question-guided exploration capability of the agent, we propose the Answer Quality-Guided Navigator (AQ-Nav). The AQ-Nav module conducts environment exploration while delivering high-relevance visual observations aligned with the target query to the VLM for open-ended question answering. To address long-horizon exploration challenges, we devise

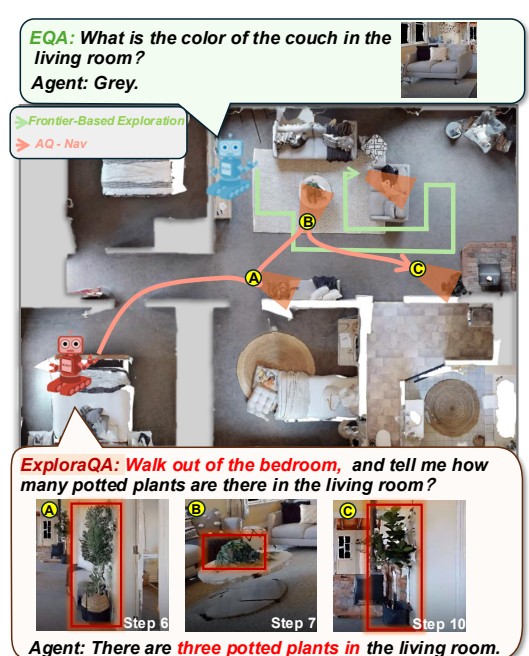

Figure 1: Comparison of ExploraQA with existing EQA datasets, highlighting long-horizon exploration and proactive navigation.

a Topology-Aware Keyframe Search Module in AQ-Nav that strategically selects critical viewpoints using topological environment maps and visual saliency analysis, reducing redundant observations and enhancing VLM processing efficiency. To optimize AQ-Nav's question-driven exploration strategy, we propose an Answer Quality Reward Mechanism (AQR): During training, we employ dual LLM evaluators—an answer VLM generates responses from observed visual sequences, while a

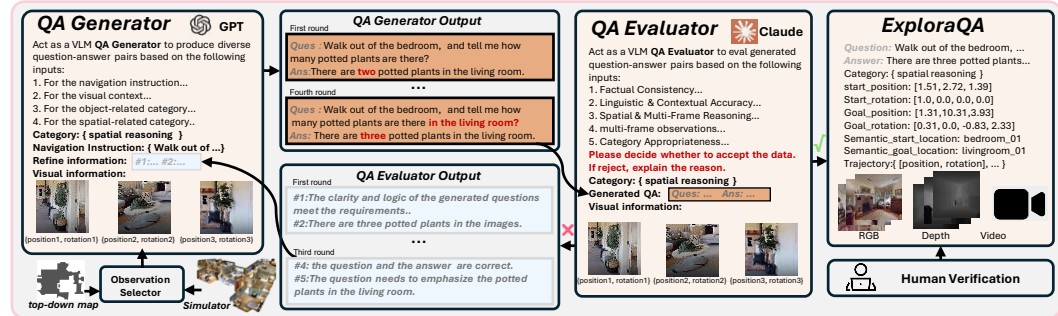

Figure 2: **The ExploraQA Construction Pipeline.** The process begins with an Observation Selector identifying optimal viewpoints from a top-down map and capturing RGB images with poses. This data, combined with navigation instructions, is sent to a QA Generator to create QA pairs. Next, a QA Evaluator assesses the pairs, retaining high-quality examples and providing corrective feedback to the generator in a refinement loop. All annotations are human-verified.

critic LLM scores answer quality based on semantic completeness and factual consistency. The AQ-Nav is optimized to maximize these scores through reinforcement learning, ensuring exploration trajectories yield maximally informative visual evidence. Experimental results demonstrate that AQ-Nav achieves 5.4% $E\_score$ absolute improvement in test unseen set of ExploraQA over the state-of-the-art VLN-based navigators when paired with identical VLMs, validating its effectiveness in question guided exploration.

## 2 RELATED WORK

### 2.1 EMBODIED QUESTION & ANSWERING

Embodied Question Answering (EQA) marks a significant paradigm shift beyond earlier Visual Question Answering (VQA) tasks Xue et al. (2024); Ji et al. (2024); Zhu et al. (2024), which focused on static, pre-acquired visual data Marino et al. (2019); Zellers et al. (2019); Li et al. (2023); Mangalam et al. (2023); Zhong et al. (2022). The EQA task is a challenging problem that emerged alongside advancements in robotics. It requires an agent to navigate and perceive a 3D environment to answer questions related to that environment. Early works primarily focused on Question Answering tasks using pre-acquired static images, Ishmam et al. (2024a;b); Marino et al. (2019), videos Zhong et al. (2022). Das et al. (2018a) introduced a synthetic dataset that involves controlling a virtual robot to navigate within an environment, gather visual information from an egocentric perspective, and answer language-based questions. Subsequent works, such as Wijmans et al. (2019); Liu et al. (2024a); Yu et al. (2019); Das et al. (2018b), designed template-based questions focusing on aspects like color and object locations. Later studies Ren et al. (2024); Majumdar et al. (2024), began exploring methods for actively searching for targets within the environment and answering corresponding questions. However, current EQA datasets remain limited by insufficient navigation information, resulting in predominantly short-range question answering tasks. This limitation highlights the need for more comprehensive benchmarks that better capture the complexities of embodied intelligence in diverse environments.

### 2.2 VISION-AND-LANGUAGE NAVIGATION

The development of instruction-following navigation agents has emerged as a prominent research area in recent years Zhang & Kordjamshidi (2024); Wu et al. (2022); Hong et al. (2024); Chen et al. (2024); Tian et al. (2024). This field was significantly advanced by the introduction of the Room-to-Room navigation task Anderson et al. (2018), which established a benchmark requiring agents to navigate between rooms based on natural language instructions. Building upon this foundation, subsequent research Liu et al. (2023); Blukis et al. (2019); Lee et al. (2024) has expanded the paradigm through diverse tasks and datasets that address various challenges in instruction-guided navigation. Notably, Qi et al. (2020) developed a framework to specifically evaluate an agent's ability to ground and execute natural language instructions, while Chen et al. (2019) extended the scope to outdoor

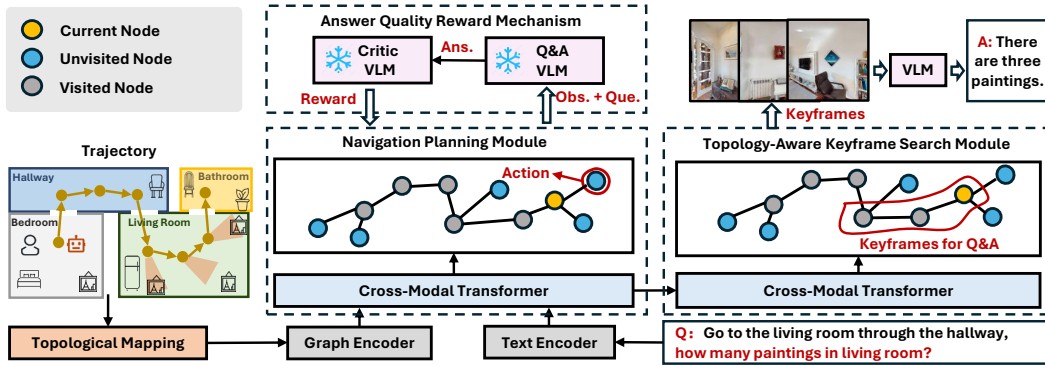

Figure 3: **Overview of our proposed method.** Our method includes a Topological Mapping Module that dynamically constructs and updates a topological map based on observational inputs. This map, along with the current question, is fed into the Navigation Planning Module, which predicts optimal exploration targets. The TAKS module subsequently identifies question-relevant visual regions to facilitate VLM question answering. We optimize the navigator using AQR Mechanism.

urban navigation scenarios with complex verbal directives. While existing works like VLN-CE Krantz et al. (2020) primarily focus on precise trajectory following, our work distinguishes itself by emphasizing the agent's active question-answering capability. This approach moves beyond passive instruction-following to enable more natural human-agent interaction during navigation tasks, where the agent can proactively seek clarification or additional information to answer the questions.

## 3 ExploraQA Dataset

We propose **Iterative EQA Data Generation Framework**, a systematic approach for constructing the ExploraQA dataset. This framework facilitates the generation of *long-horizon exploration* and *proactive navigation trajectories*. Additionally, we augment the dataset with *comprehensive viewpoint annotations* corresponding to each question, enhancing its utility for EQA research.

### 3.1 Iterative EQA Data Generation Framework

Our data is collected in the Habitat environment Savva et al. (2019). While recent studies Han et al. (2025); Patel et al. (2024); Zhou et al. (2025) have leverages VLMs for direct question-answer pair generation, existing VLMs Li et al. (2024; 2025); Wu et al. (2024) are not readily applicable to generating the EQA dataset featuring long-horizon exploration and proactive trajectory. To overcome this limitation, we introduce an Iterative EQA Data Generation Framework as shown in Fig. 2. The framework is composed of three synergistic components:

**Observation Selector.** To generate high-quality visual data aligned with proactive trajectories, we propose an observation selection strategy that identifies optimal viewpoints near the trajectory endpoints in VLN-CE Krantz et al. (2020). The geometric plausibility of these viewpoints is verified through a 2D occupancy map and heuristic-based filtering, enforcing three critical constraints: i) The pose must be within 3 meters of the trajectory endpoint to keep relevant content in view. ii) The viewpoint must not directly face a wall, ensuring the captured images contain sufficient visual information. iii) The relative orientation between poses is managed to guarantee non-overlapping visual coverage. Crucially, as each trajectory is paired with navigation instructions (e.g., "walk out of the bedroom..."), the selected viewpoints inherently align with the described navigation context.

**QA Generator.** We propose a QA Generator that leverages the capabilities of GPT-4o Hurst et al. (2024) to create proactive QA pairs. The generator produces an initial QA pair based on three inputs: i) the visual input captured from *Observation Selector*. ii) refinement feedback from *QA Evaluator*. iii) navigation instruction from VLN-CE Anderson et al. (2018). Through this process, it generates proactive QA pairs.

**QA Evaluator.** To validate the generated data and provide effective feedback for quality improvement, we employ an Evaluator based on Claude Sonnet 3.5 Anthropic (2024) to assess data quality

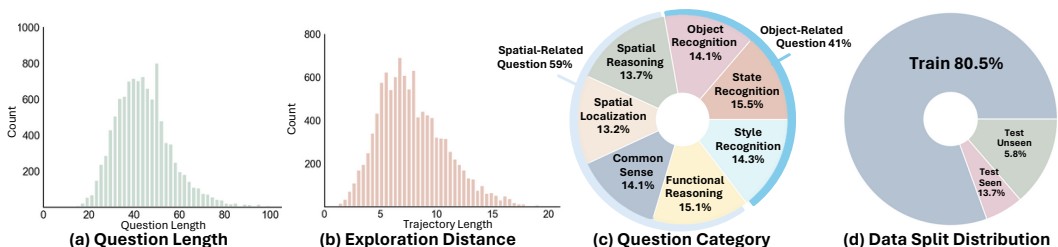

Figure 4: **Overview of our dataset statistical analysis.** (a) - (d) Statistical analysis of the dataset, covering question lengths, exploration distances, question categories, and dataset splits.

and generate refinement feedback. The evaluator examines the generated QA pairs through multiple criteria to determine their acceptance, employing the following strategies: i) To ensure QA pairs are proactive, the evaluator assesses the appropriateness of the guidance instruction of each answer. ii) To minimize factual errors, the evaluator verifies whether all objects mentioned in the question are present in the visual input. iii) To ensure accurate spatial relationships, the evaluator checks whether answers are properly derived from the multi-frame data. Finally, the evaluator consolidates all assessment results to determine whether to accept the data or proceed with refinement. If refinement is needed, the Explainer generates refinement feedback based on the Evaluator's findings and incorporates it into the refinement history for the QA generator to improve subsequent iterations.

## 3.2 DATA QUANTITATIVE ANALYSIS

This section analyzes the ExploraQA dataset in terms of question categories and data splits.

**Category Analysis.** As illustrated in Fig. 4 (a), we collected 12,436 QA pairs with corresponding trajectories, featuring an average question length of 44 words. Fig. 4 (b) demonstrates our trajectory distribution, with an average length of 10.4 meters, the extended trajectories pose challenges for the model to reach question-relevant regions. We designed seven distinct question types as shown in Fig. 4 (c). The questions are categorized into two primary types: object-related and spatial-related.

**Dataset Split.** As shown in Fig. 4 (d), we divide the dataset into subsets: Train, Test Seen, and Test Unseen. The word 'Seen' means the visual environments that have been seen in the train separation. We assign 72 scenes for training, where the train set contains 9,819 question-answer pairs and Test Seen set contains 778 QA pairs. The Test Unseen is assigned 8 scenes with 1,839 QA pairs.

## 4 THE ANSWER QUALITY-GUIDED NAVIGATOR

There are three core components in AQ-Nav: i) the topological map constructor that builds environmental structural graphs, ii) a navigation planning module that predicts subgoals based on topological graphs, and iii) a Topology-Aware Keyframe Search (TAKS) module that locates question-relevant keyframes using environmental cues.

### 4.1 TASK DEFINITION

EQA requires an intelligent agent to answer a natural language question $Q$ through active exploration in a 3D environment $S$. The agent is initialized at pose $P_0 = (x, y, z, h)$, where $(x, y, z)$ denotes its spatial coordinates and $h$ represents its heading orientation. At each timestep $t$, the agent selects a navigational action $A_t$ (e.g., MOVEFORWARD, TURNLEFT/RIGHT, or STOP), transitioning to a new pose $P_t$. The episode terminates upon executing the STOP action, after which the agent generates its final answer based on the accumulated observation history $O = \{o_1, o_2, ..., o_t\}$.

### 4.2 TOPOLOGICAL-BASED NAVIGATOR

Our navigator uses a two-stage process: a Topological Map Conductor dynamically builds a graph map, which a Navigation Planning Module then uses with text instructions to select waypoints. This graph-based approach also inherently enables the agent to backtrack.

**Topological Map Conductor.** Topological Map Conductor presents an online topological mapping approach for autonomous navigation that dynamically constructs a graph-structured map $\mathcal{G}_t = \langle \mathcal{N}_t, \mathcal{E}_t \rangle$ at step $t$ from visual observations. The agent predicts potential waypoints using the current depth observations, then integrates them to update $\mathcal{G}_t$ to $\mathcal{G}_{t+1}$ An et al. (2024). The graph $G_t$ comprises nodes that store both positional coordinates for graph integration and their associated visual features, with edges representing reachability and pairwise distances between nodes. The $\mathcal{N}_t$ has two subsets, *i.e.*, visited nodes $\mathcal{N}_t^v$ and unvisited nodes $\mathcal{N}_t^u$. Nodes in $n_{t,i}^v \in \mathcal{N}_t^v$ represent previously traversed locations in the navigation trajectory, each storing visual features $\boldsymbol{I}_{t,i}^v \in \mathbb{R}^{12 \times D}$, which encoded 12 distinct directional observations at the respective position. In contrast, $n_{t,i}^u \in \mathcal{N}_t^u$ denote unexplored locations, preserving visual features of their directions from the latest adjacent node $\boldsymbol{I}_{t,i}^u \in \mathbb{R}^{1 \times D}$. To navigate to a nearby unvisited node on $\mathcal{G}_t$, the agent computes the relative direction and executes a move-forward action. By constructing $\mathcal{G}_t$, the continuous navigation task is transformed into a discrete graph-based navigation problem, where action prediction reduces to selecting the next unvisited node to explore.

**Navigation Planning Module.** This module leverages textual information to decide which unvisited node to visit on $\mathcal{G}_t$. The module consists of a multimodal transformer Tan & Bansal (2019), with two parallel encoder dedicated to processing textual inputs and the graph structure $G_t$, respectively. The text encoder, which is a transformer first processes the instruction into contextual word embeddings $\boldsymbol{X}$. To encode $\mathcal{G}_t$, we first encode the topological structure of the graph and then fuse it with visual features. For brevity, we omit the time subscript $t$ in the subsequent discussion of this section. For topological encoding, the coordinate and visiting time step of each node are embedded into a vector, which is further processed via Graph-Aware Self-Attention (GASA) An et al. (2024) encoding along with $\mathcal{E}$ to obtain the topological features $\{\boldsymbol{\tau}_k\}_{k=1}^K$, where $K$ is the node number of $\mathcal{N}$. For visual features, we average the omnidirectional visual features in $\mathcal{N}^v$ to ensure consistent dimensionality for each node, denoted as $\{\hat{\boldsymbol{I}}_k\}_{k=1}^K$. Subsequently, the feature representation of $\mathcal{G}_t$ is represented as:

$$\{\boldsymbol{\rho_k}\}_{k=1}^K = \{\hat{\boldsymbol{I}}_k + \boldsymbol{\tau}_k\}_{k=1}^K. \tag{1}$$

Subsequently, multi-layer cross-attention is applied between $\boldsymbol{X}$ and $\{\boldsymbol{\rho_k}\}_{k=1}^K$, producing the updated feature $\{\hat{\boldsymbol{\rho}_k}\}_{k=1}^K$. The features in $\{\hat{\boldsymbol{\rho}_k}\}_{k=1}^K$ belonging to $\mathcal{N}^u$ are concatenated to form $\hat{\boldsymbol{\rho}}^v \in \mathbb{R}^{K_v \times D}$. The $\hat{\boldsymbol{\rho}}^v$ are then scored via a feed-forward network to predict the action distribution $a$ :

$$a = \text{softmax}(\text{FFN}([\boldsymbol{s}_0, \hat{\boldsymbol{\rho}}^v])), \tag{2}$$

where $\boldsymbol{s}_0 \in \mathbb{R}^{1 \times D}$ is a learnable embedding indicating stop action. The agent selects the unvisited node (or stop action) with the highest probability in $a$ as the current action $A_t$.

### 4.3 TOPOLOGY-AWARE KEYFRAME SEARCH MODULE

In the ExploraQA task, agents must perform long-range exploration where question-relevant visual information is sparsely located. This characteristic poses a problem: providing the entire observation history to a VLM causes significant hallucinations due to a low signal-to-noise ratio. To overcome this, we introduce the TAKS module. This method filters the navigation history by analyzing frames from all visited locations to select a concise set of topologically-relevant keyframes.

As $\{\boldsymbol{\rho_k}\}_{k=1}^K$ encodes the complete trajectory topology, positional relationships, and visual features, we use it to predict the locations of question-relevant keyframes. Since nodes in $\mathcal{N}^u$ lack observational data at their corresponding locations, we exclusively select keyframes from $\mathcal{N}^v$. We concatenate the features in $\{\boldsymbol{\rho_k}\}_{k=1}^K$ that belong to $\mathcal{N}^v$, denoted as $\boldsymbol{\rho}^v$. $\boldsymbol{\rho}^v$ are then processed through 2 transformer layers. The final selection scores $v$ are computed via a two-layer FFN. We select nodes with confidence scores in $v$ exceeding $\beta$ as question-relevant locations, and use the agent's observations at these positions as keyframes.

### 4.4 ANSWER-QUALITY GUIDED TRAINING PIPELINE

To enhance the agent's combined navigation and question-answering capabilities, we designed a three-phase training strategy. This process begins by establishing foundational navigation ability via IL, followed by fine-tuning with a hybrid RL approach, and concludes with a specialized optimization phase to improve keyframe selection quality.

Table 2: Comparison across different navigators on the ExploraQA with QwenVL2.5-7B

| Split | Method | Object | | | Spatial | | | Average | |
|---|---|---|---|---|---|---|---|---|---|
| | | $C\_score$ | TL | $E\_score$ | $C\_score$ | TL | $E\_score$ | $C\_score$ | $E\_score$ |
| Test Seen | Random | 28.5 | 29.5 | 20.1 | 27.7 | 32.9 | 18.6 | 28.1 | 19.3 |
| | Seq2Seq | 36.7 | 12.7 | 29.8 | 35.7 | 11.3 | 30.0 | 36.2 | 29.9 |
| | RecBERT | 39.4 | 12.3 | 31.7 | 36.6 | 12.3 | 31.5 | 37.9 | 31.6 |
| | ETPNav | 42.6 | 11.6 | 35.1 | 42.6 | 11.3 | 35.6 | 42.6 | 35.3 |
| | Ours | **48.5** | **10.8** | **42.4** | **45.1** | **10.6** | **39.3** | **46.7** | **40.7** |
| Test Unseen | Random | 29.2 | 33.4 | 18.9 | 28.9 | 35.3 | 19.4 | 28.9 | 19.4 |
| | Seq2Seq | 33.8 | 12.5 | 26.7 | 32.3 | 13.0 | 25.3 | 33.0 | 26.0 |
| | RecBERT | 36.5 | 12.2 | 29.4 | 35.0 | 12.6 | 30.7 | 35.7 | 30.1 |
| | ETPNav | 41.2 | 11.9 | 32.7 | 41.2 | 12.6 | 32.4 | 41.2 | 32.5 |
| | Ours | **44.9** | **10.8** | **38.2** | **45.8** | **11.0** | **37.6** | **45.4** | **37.9** |

**Phase 1: Navigator Pre-Training.** We pre-train the navigation model on expert trajectories using IL. The core objective is to equip the agent with stable and general-purpose navigation ability.

**Phase 2: Navigator Fine-Tuning.** We fine-tune the agent using a hybrid approach that combines IL and RL. The IL component ensures the model maintains its foundational navigation capabilities, while the RL component uses two core reward mechanisms to guide question-oriented exploration: *(i) Answer-Quality Reward.* This reward quantifies the value of a keyframe selected at the navigation endpoint for answering a given question. A QA VLM (QwenVL2.5-7B) generates a candidate answer, which is then assessed by a distinct Critic VLM (InternVL2.5-8B) on a 1-5 scale. To mitigate reward hacking and account for VLM stochasticity, we adopt a stringent policy: a positive reward $R_{AQR} = 1$ is granted only for a perfect score of 5, and 0 otherwise. *(ii) Navigation Success Reward.* The agent receives $R_{Target} = 1$ if its final position is within 3 meters of the target destination, and 0 otherwise. The final PPO reward is a weighted combination:

$$R = \lambda \cdot R_{\text{AQR}} + (1 - \lambda) \cdot R_{\text{Target}}. \tag{3}$$

This combined signal jointly optimizes exploration strategy and keyframe selection in the TAKS module.

**Phase 3: TAKS Module Fine-tuning.** We freeze the navigator's parameters and fine-tune only the TAKS module. The ground-truth is defined by the semantic context of the target location (e.g., "Bedroom A"). We then use a MSE loss to align the module's predicted viewpoint importance scores with the ground-truth relevance labels. This targeted optimization enhances the model's ability to select the most crucial viewpoints for answering questions.

## 5 EXPERIMENTS

To evaluate ExploraQA, our framework integrates navigation models with VLMs to assess agents based on their answers. We selected multiple baseline methods and state-of-the-art (SOTA) models from related benchmarks, and conducted comprehensive evaluation on ExploraQA.

**Implementation Details.** We train our model on two RTX 4090 GPUs for a total of approximately 60 hours, using a batch size of 16 and learning rate of 1e-5. The training process is divided into three distinct phases. First, the navigator undergoes pre-training with IL for 15,000 epochs. Second, the navigator is fine-tuned for 5,000 epochs using a combination of IL and RL at a 3:1 ratio, with the RL coefficient $\lambda$ set to 0.25. In the final stage, all navigator parameters are frozen, and only the TAKS module is trained for 5,000 epochs, with the confidence threshold $\beta$ set to 0.5.

**Comparison Baselines.** We evaluate our approach against several navigation baselines on the ExploraQA task, using QwenVL2.5-7B Bai et al. (2023) for visual question answering, which is distinct from GPT-4o Hurst et al. (2024) to avoid reward hacking. Baselines include: (1) Random Policy; (2) Seq2Seq Krantz et al. (2020); (3) RecBERT Hong et al. (2022); (4) ETPNav An et al. (2024).

**Evaluation Metrics.** We evaluate the model using two key metrics: question correctness score ($C\_score$) and completion efficiency score ($E\_score$). For $C\_score$ assessment, following the

Table 3: Ablation study on the TAKS module and the AQR on ExploraQA.

| Split | AQR | TAKS | Object | | | Spatial | | | Average | |
|---|---|---|---|---|---|---|---|---|---|---|
| | | | $C\_score$ | TL | $E\_score$ | $C\_score$ | TL | $E\_score$ | $C\_score$ | $E\_score$ |
| Test Seen | ✗ | ✗ | 42.6 | 11.6 | 35.1 | 42.6 | 11.3 | 35.6 | 42.6 | 35.3 |
| | ✓ | ✗ | 44.9 | **10.8** | 39.0 | 42.8 | **10.6** | 37.3 | 43.7 | 38.1 |
| | ✓ | ✓ | **48.5** | 10.8 | **42.4** | **45.1** | 10.6 | **39.3** | **46.7** | **40.7** |
| Test Unseen | ✗ | ✗ | 41.2 | 11.9 | 32.7 | 41.2 | 12.6 | 32.4 | 41.2 | 32.5 |
| | ✓ | ✗ | 42.3 | **10.8** | 35.2 | 43.4 | **11.0** | 35.8 | 42.9 | 35.5 |
| | ✓ | ✓ | **44.9** | 10.8 | **38.2** | **45.8** | 11.0 | **37.6** | **45.4** | **37.9** |

methodology in Liu et al. (2024b), we employ an LLM to evaluate the model's responses on a 5-point scale for each question, where 1 represents a completely incorrect answer and 5 indicates a fully correct response. Intermediate scores $\sigma$ (2–4) correspond to varying levels of correctness relative to the reference answer. The final $C\_score$ metric is then computed as:

$$C\_score = \frac{\sigma - 1}{4} \times 100\%. \tag{4}$$

We introduce the $E\_score$, a unified metric that comprehensively evaluates both the question-answering capability and exploration efficiency of the model, which is defined as:

$$E\_score = \frac{\sigma - 1}{4} \times \frac{l}{\max(p, l)} \times 100\%, \tag{5}$$

where $p$ denotes the trajectory length taken by the agent, and $l$ represents the reference path length.

## 5.1 QUANTITATIVE RESULT

We evaluate the EQA performance of models using two primary metrics: $E\_score$ and $C\_score$.

**Comparison with Baselines.** As illustrated in Tab. 4.4, our method achieves significant improvements over the ETPNav An et al. (2024) baseline. Specifically, the proposed approach elevates the $C\_score$ by 4.1% on test seen environment, indicating that the critical information captured by our method effectively enhances question-answering capabilities. Moreover, our method reduces the navigation distance by 0.8m and 0.7m for object-related and spatial-related questions, respectively. Collectively, these enhancements in both question answering and navigation efficiency lead to a 5.4% increase in $E\_score$ over ETPNav in seen environments. These advancements stem from our TAKS module and AQR mechanism. While random policies achieve an $E\_score$ of 19.3%, they significantly underperform compared to our method's 40.7%. Indicating that achieving the ExploraQA tasks without understanding instructions and visual information is extremely difficult.

**Performance on Object and Spatial Related Questions.** For object-related questions, which primarily test the navigation model's ability to locate relevant objects, our method achieves performance improvements of 7.3% in $E\_score$ and 5.9% in $C\_score$ compared to ETPNav in seen environments, demonstrating enhanced capability for target object localization. For spatial-related questions, which require the model to localize multiple spatial positions, the performance improvement is 3.7%, indicating our approach's superior multi-target exploration capabilities. Spatial-related questions consistently exhibit lower performance than object-related questions, suggesting that the dataset's spatial reasoning tasks present greater challenges, as they necessitate accurate identification of multiple spatial locations to correctly answer queries.

**Generalization on Unseen Cases.** As shown in Tab. 4.4, our method achieves $E\_score$ metrics of 38.2% and 37.6% for object-related and spatial-related questions respectively, demonstrating robust adaptability to novel scenarios in unseen test environments. While pure imitation learning is constrained by the quality of expert demonstrations, our hybrid methodology enables the agent to discover more optimal policies through reinforcement learning mechanisms. Concurrently, the average exploration distance decreased by 1.1m for object-related and 1.6m for spatial-related questions, indicating enhanced navigation efficiency through optimized trajectory planning and improved target localization precision.

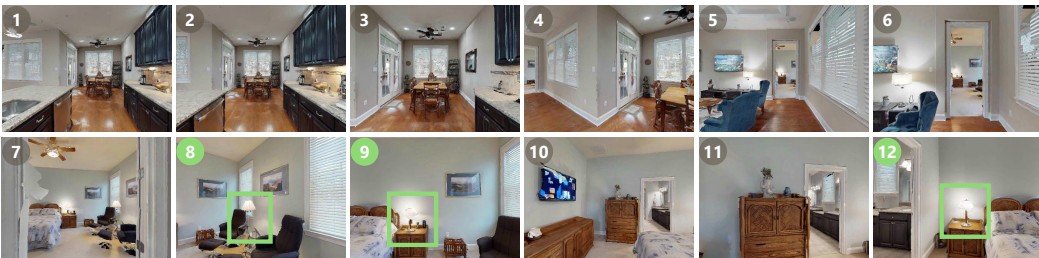

Question: Please first go forward to the dining table and then turn left into the room... Where are the three table lamps located in the room?
Correct Answer: Two table lamps are on the nightstand, and a third one is placed between two black chairs.
Predict Answer: I can see three table lamps in the bedroom, with two on the nightstand and another one between two black chairs.

Figure 5: **A visualization example from the ExploraQA test set.** AQ-Nav can accurately localize the target objects relevant to the given questions with TAKS.

## 5.2 ABLATION STUDY.

In this section, we conduct an ablation study on the two methods we proposed, AQR and TAKS, to show their effectiveness.

**Ablation Study of AQR.** As illustrated in Tab. 5.1, the application of AQR finetuning yields substantial improvements in EQA performance, with the $C\_score$ increasing by 1.1% and 1.7% on the seen and unseen sets, respectively. Furthermore, the navigation distance decreases significantly, leading to $E\_score$ improvements of 2.8% on the seen set and 3.0% on the unseen set, indicating that the AQR reward mechanism successfully minimizes trajectory length and enhances overall navigation efficiency.

**Ablation Study of TAKS.** Applying the TAKS module, our model exhibits notable improvements in object-related questions, achieving gains of 3.4% and 3.0% on seen and unseen environments, respectively. This suggests that when the agent encounters objects, the topological interest module effectively leverages cross-modal vision-language topological information to identify question-relevant regions. Furthermore, for spatial-related questions, performance improves by 4.6% and 5.2% on seen and unseen environments, respectively. This demonstrates that the TAKS module remains robust in selecting key observation points even in complex multi-target environments.

## 5.3 QUALITATIVE RESULT

As shown in Fig. 5, given the language instruction, the model first correctly turns left and enters the room. During exploration, it successfully locates the first and second lamps, but temporarily loses visual tracking of them while scanning other areas. After detecting the final lamp, the exploration phase concludes. The topological interest module then analyzes the topological map and vision-language features, identifying step 8, step 9, and step 12 as key frames. This approach enables the model to filter redundant visual information in long trajectories and select optimal observations for question answering.

## 6 CONCLUSION

In this paper, we present ExploraQA, a novel benchmark for evaluating EQA agents on long-horizon exploration and open ended question-answering ability. The benchmark encompasses 7 distinct embodied question types and features extended exploration trajectories. To build ExploraQA, we propose an iterative VLM-based pipeline for question-answer generation. To address the requirement for proactive exploration in EQA agents, we propose AQ-Nav, which leverages VLMs to assess exploration trajectories and select key frames for decision-making. The superior EQA performance achieved by AQ-Nav demonstrates the effectiveness of the proposed approach. ExploraQA introduces a more challenging benchmark in the EQA domain, aiming to advance research in this field.

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

# A   APPENDIX

## A.1   VLM-AS-A-JUDGE HUMAN ALIGNMENT

The evaluation of open-vocabulary responses to inquiries, particularly within the domain of question answering, presents a significant challenge in the field of artificial intelligence. Traditional human evaluation, while considered the gold standard, is both cost-prohibitive and time-intensive. Consequently, the adoption of automated evaluation metrics is imperative for robust and efficient benchmarking. In this study, we leverage a VLM based automated evaluation metric, designated as VLM-Match.

To validate the efficacy of this metric, we focus of our investigation was to determine the degree of concordance between the VLM-Match metric and human evaluators. To this end, we designed an experiment to measure this alignment. A random sample of 300 questions was drawn from the exploraQA dataset and subsequently answered by our SOTA model. These 300 responses were then independently assessed by three human annotators, who followed an evaluation rubric analogous to that used by the VLM-Match metric.

Our analysis reveals a strong positive correlation between the human evaluations and the VLM-based assessments. As shown in Tab. 4. The Spearman's rank correlation coefficient ($\rho$) between the aggregated human judgments and the VLM-Match scores was 0.909. This result indicates a high degree of consistency with human judgment.

Table 4: **Per-annotator Spearman-$\rho$.** The results indicate excellent agreement between human raters and also between human and LLM scoring methodologies.

| annotator | vs. Other Humans | vs. LLM |
|:---:|:---:|:---:|
| 1 | 0.91 | 0.91 |
| 2 | 0.92 | 0.90 |
| 3 | 0.91 | 0.91 |

## A.2   COMPARISON BASELINES.

The navigation baselines include: (1) Random Policy: The agent selects actions uniformly at random to navigate through the environment. (2) Seq2Seq Baseline Krantz et al. (2020): A sequence-to-sequence model that encodes visual observations and language instructions using a transformer encoder, then decodes the action sequence in an autoregressive manner. (3) RecBERT Baseline Hong et al. (2022): A candidate waypoints predictor that generates accessible waypoints from visual observations, then a high-level navigation agents VLN-BERT to make decisions. (4) ETPNav Baseline An et al. (2024): A topology-aware navigation method that dynamically constructs and updates a navigation graph to guide the agent toward the target location.

## A.3   NAVIGATION RESULTS

While effective navigation is a critical component of the ExploraQA task, our primary evaluation centers on the quality of the embodied question answering process. Consequently, we prioritize the task-specific performance metrics, $E\_score$ and $C\_score$, over Vision-and-Language Navigation (VLN) performance indicators. These scores are designed to directly measure the effectiveness of the agent's exploration and the accuracy of its final answer. Nevertheless, we also report standard VLN performance metrics in Tab. 5.

Table 5: **VLN Metrics Comparison.** Our method also demonstrates SOTA performance under VLN metrics.

| Split | Method | NE | OSR | SR | SPL |
|---|---|---|---|---|---|
| Test Seen | Seq2Seq | 7.19 | 42.5 | 35.9 | 31.5 |
| | RecBERT | 5.11 | 57.3 | 49.1 | 42.4 |
| | ETPNav | 3.926 | 71.5 | 63.8 | 56.7 |
| | Ours | **3.767** | **71.2** | **66.1** | **60.0** |
| Test Unseen | Seq2Seq | 7.387 | 39.8 | 31.5 | 26.3 |
| | RecBERT | 5.92 | 49.3 | 38.9 | 34.2 |
| | ETPNav | 5.272 | 57.8 | 51.6 | 43.7 |
| | Ours | **5.144** | **61.6** | **52.2** | **45.1** |

Table 6: **Category-level $C\_score$ Performance on ExploraQA.** Comparison of $C\_score$ across different navigators on the ExploraQA benchmark using QwenVL2.5-7B.

| Split | Method | Style Rec. | Object Rec. | State Rec. | Spatial Loc. | Func. Reas. | Common Sense | Spatial Reas. |
|---|---|---|---|---|---|---|---|---|
| Seen | Seq2Seq | 39.7 | 34.1 | 35.8 | 37.2 | 42.0 | 24.3 | 24.7 |
| | RecBERT | 42.1 | 36.7 | 39.1 | 40.1 | 43.9 | 26.8 | 31.4 |
| | ETPNav | 45.5 | 39.5 | 42.5 | 45.8 | 47.7 | 31.6 | 41.3 |
| | **Ours** | **50.7** | **42.5** | **52.5** | **45.5** | **50.3** | **39.0** | **42.8** |
| Unseen | Seq2Seq | 30.4 | 31.3 | 39.0 | 31.5 | 35.7 | 30.9 | 31.0 |
| | RecBERT | 34.4 | 33.3 | 41.1 | 35.8 | 38.4 | 32.1 | 33.8 |
| | ETPNav | 38.8 | 38.1 | 46.1 | 44.8 | 44.2 | 37.5 | 39.1 |
| | **Ours** | **42.1** | **41.7** | **50.3** | **45.2** | **48.2** | **47.8** | **42.2** |

### A.4 CATEGORY-LEVEL PERFORMANCE ON EXPLORAQA

To offer a comprehensive evaluation, we present a detailed metrics across each of the seven question categories. We report the exploration effectiveness, measured by the $C\_score$, in Tab. 7. The final answer accuracy, measured by the $E\_score$, is presented in Tab. 6.

### A.5 DATA GENERATION DETAILS

The prompt used for the QA Generator is detailed in Fig. 6 and the prompt used for the QA Evaluator is detailed in Fig. 7.

**Human Verification.** To ensure the quality of our dataset, all annotations underwent a rigorous human verification process. We initially employed VLMs to generate preliminary labels. These candidates were then reviewed by human annotators to confirm semantic correctness, visual alignment, and adherence to formatting standards. Our protocol dictated that each entry be evaluated by three annotators; an entry that failed to secure unanimous approval was flagged. In this process, 83% of the labels were approved directly. The remaining 17% were manually revised by our annotators and subsequently passed a second round of verification before their final inclusion in the dataset.

### A.6 DATA QUALITATIVE ANALYSIS

To advance the development of EQA, we propose the ExploraQA dataset to address the limitations of existing EQA benchmarks. **First,** while current EQA datasets primarily focus on object-centric questions (e.g., "What color is this chair?"), we extend the scope by introducing spatial-related questions alongside traditional object-related queries. **Second,** existing EQA datasets lack navigation-aware question design, where methods like forward-only policies in Wijmans et al. (2019) or frontier-based exploration in Majumdar et al. (2024) suffice for task completion. To bridge this gap, we leverage the powerful linguistic capabilities of GPT-4o Hurst et al. (2024) and Claude Sonnet 3.5 Anthropic (2024), we generate questions that naturally incorporate navigation instructions from Krantz et al. (2020), thereby constructing a more realistic and challenging benchmark.

Table 7: **Category-level *E_score* Performance on ExploraQA.** Comparison of $E\_score$ across different navigators on the ExploraQA benchmark using QwenVL2.5-7B.

| Split | Method | Style Rec. | Object Rec. | State Rec. | Spatial Loc. | Func. Reas. | Common Sense | Spatial Reas. |
|---|---|---|---|---|---|---|---|---|
| Seen | Seq2Seq | 32.6 | 27.3 | 29.2 | 33.3 | 35.0 | 20.0 | 27.8 |
| | RecBERT | 33.6 | 29.3 | 32.1 | 35.0 | 36.9 | 22.5 | 27.9 |
| | ETPNav | 37.4 | 33.1 | 34.3 | 41.4 | 39.0 | 26.6 | 32.3 |
| | **Ours** | **44.5** | **36.3** | **46.6** | **41.0** | **43.9** | **34.2** | **35.7** |
| Unseen | Seq2Seq | 23.7 | 24.1 | 31.7 | 24.7 | 28.8 | 23.2 | 24.4 |
| | RecBERT | 27.7 | 26.3 | 33.8 | 28.1 | 32.1 | 27.8 | 33.8 |
| | ETPNav | 29.5 | 29.2 | 38.6 | 35.6 | 35.3 | 28.1 | 31.2 |
| | **Ours** | **35.5** | **35.4** | **43.1** | **36.5** | **40.2** | **38.8** | **35.0** |

Table 8: **Ablation study.** The coefficient $\lambda$ between the two reinforcement learning objectives in the navigation model.

| $\lambda$ | Object | | | Spatial | | | Average | |
|---|---|---|---|---|---|---|---|---|
| | $C\_score$ | TL | $E\_score$ | $C\_score$ | TL | $E\_score$ | $C\_score$ | $E\_score$ |
| 0.00 | 44.0 | 11.5 | 35.9 | 44.8 | 11.8 | 35.2 | 44.4 | 35.5 |
| 0.25 | **44.9** | **10.8** | **38.2** | **45.8** | **11.0** | **37.6** | **45.4** | **37.9** |
| 0.50 | 42.8 | 11.3 | 35.1 | 44.4 | 11.5 | 35.3 | 43.7 | 35.2 |
| 0.75 | 41.6 | 12.1 | 33.5 | 42.7 | 13.0 | 32.7 | 42.2 | 33.1 |
| 1.00 | 36.3 | 14.8 | 27.5 | 36.6 | 15.0 | 27.2 | 36.5 | 27.3 |

As illustrated in Fig. 8, the ExploraQA dataset incorporates explicit navigation instructions, enabling an explore-then-answer paradigm. The object-related questions are categorized into three types: style recognition, object recognition, and state recognition Additionally, Fig. 9 presents our spatial-related questions, which encompass four challenging subtypes: spatial localization, spatial understanding, functional reasoning and commonsense reasoning.

## A.7 AQR MECHANISM DETAILS

The AQR mechanism is designed to generate appropriate rewards for model optimization. During the QA process, we employ the same prompt structure as used in the final VQA evaluation. As illustrated in Fig. 10, the prompt includes information from multiple images along with their corresponding pose, which aids the model in understanding spatial relationships across the images. For scoring, we adopt a prompt design similar to Majumdar et al. (2024) as shown in Fig. 11, consistency with the evaluation phase. Specifically, we utilize a few-shot learning approach to help the model discern the meaning behind different score levels.

## A.8 ABLATION STUDY

We investigate the effect of different reward component ratio, with the ablation results presented in Tab. 8. The experiments demonstrate that the optimal performance is attained when $\lambda = 0.25$ Compared to using only the target reward $\lambda = 0$, our approach achieves improvements of 1.1% in $C\_score$ and 2.4% in $E\_score$ on the unseen set, validating the effectiveness of the hybrid reward design specifically for the EQA task.

## A.9 DETAILS OF LLM USAGE

We used large language models solely to assist with language polishing, including grammar correction, improving sentence fluency, and refining the clarity of exposition. The research ideas, methods, analyses, and experimental results were fully developed and validated by the authors.

**Q&A Generator Prompt:**

Act as a Vision-Language Model **QA Generator** to produce diverse question-answer pairs based on the following inputs:

1. For the navigation instruction, you have now accurately reached the destination position according to the instructions.

2. For the visual context, it consists of 3 images captured from suitable observation positions within a 3m range of the

destination.

3. For the historical information, it is provided by your explainer based on the results of the previous round to iteratively optimize

the question-answer pairs you propose.

4. For the object-related category, the designed questions should primarily inquire about object-related information, including:

{object recognition, state recognition, style recognition}.

5. For the spatial-related category, the designed questions should consider the relationships between multiple images, including

{spatial reasoning, spatial localization, functional reasoning, commonsense reasoning}.

Please generate the question-answer pair. Note that the generated question should include the original navigation information

to ensure the agent can explore and answer the question based on it.

Input information：

**Category:** {category result}

**Visual Context:** [ {img1}, {img2}, {img3} ] (Relative positions: img1 (x1, y1, z1, h1), img2 (x2, y2, z2, h2), img3 (x3, y3, z3, h3))

**Navigation Instruction:** {navigation instruction}

**Refine History:** {refine history}

The final output should maintain the following template format:

"Question"："Question result"

"Answer"："Answer result"

Figure 6: The GPT-4o prompt for QA generator.

**Q&A Evaluator and Explainer Prompt:**

Act as a Vision-Language Model **QA Evaluator** to eval generated question-answer pairs based on the following inputs:

Evaluation Criteria:

1. Factual Consistency: Verify all objects/attributes mentioned in the QA pair exist in the visual context.

2. Linguistic & Contextual Accuracy: Check grammar, clarity, and logical coherence of the QA pair.

3. Spatial & Multi-Frame Reasoning (if applicable):

4. For spatial-related categories. Confirm the answer leverages multi-frame observations when needed.

5. Category Appropriateness: Ensure the QA pair matches the claimed category

Please analyze all the provided information for the question-answer pairs and decide whether to accept the data.

Input Information:

**Category:** {category result}

**Visual Context:** [ {img1}, {img2}, {img3} ] (Relative positions: img1 (x1, y1, z1, h1), img2 (x2, y2, z2, h2), img3 (x3, y3, z3, h3))

**Question:**{question result}

**Answer:**{answer result}

The final output should maintain the following template format:

"evaluation result": "evaluation result" (accept or reject)

**If rejected, continue to use the explainer for explanation:**

Act as a Vision-Language Model **Explainer** to eval question-answer pairs based on the following inputs:

Please provide reasoning on how to offer effective suggestions as refine history for improving data quality based on all the

given information.

The final output should maintain the following template format:

"refine history": "refine history"

Figure 7: The Claude Sonnet 3.5 prompt for QA Evaluator

**Object-Related Questions:**

**Style Recognition**

Question: To reach your destination, enter the hallway and take a left. Continue straight ahead, then make a right at the end of the hall. Enter the room with the bear and walk straight across that room into the hallway. The bedroom will be the first door on your right, stop in the doorway once you arrive. While you're there, can you tell me what the color of the curtain?

Answer: The curtain is red.

**Object Recognition**

Question: To get to your destination, go straight into the garage and then turn right into the room. After that, turn right again to enter the bathroom and wait near the sink. While you're there, can you tell me what is located on the bathroom counter?

Answer : There is a sink on the bathroom counter.

**State Recognitation**

Question: Exit the room and head down the hallway. Stop in front of the two chairs on your left. While you're there, could you check if the sliding door to the balcony is open or closed?

Answer: The balcony door is closed.

Figure 8: Illustration of the object-related questions in ExploraQA.

**Spatial-Related Questions:**

**Spatial Localization**

Question: Walk past the mirror to your right and continue through the bedroom until you reach the patio door. Wait just outside the door. By the way, do you know where the outdoor grill is located?

Answer: In the outdoor patio area with tiled flooring.

**Sptial Reasoning**

Question: Begin by going up four stairs. Once you reach the top, head towards the sunset painting. From there, make your way to the marble table, then to the oven, and finally to the dining room table. As you approach the dining table, take a moment to look to your right. What do you see between the dining table and the window?

Answer: Between the dining table and the window, there is a chair.

**Functional Reasoning**

Question: To get to the living room, go around the stairs and enter the hallway. Walk straight down the hallway until you reach the end, then take a right. You'll find yourself at the entrance to the living room. And tell me how many can sit in this room.

Answer: The room can seat four people.

**Common Sense:**

Question: To get to the bedroom, turn around and walk past the bed. Exit through the door and take a right. Enter the next door on your right, and you'll find a cozy space with a snowman quilt on the bed. Once you arrive, you might wonder: Can I stay warm in this bedroom during the winter?

Answer: Yes, there's a fireplace and heater.

Figure 9: Illustration of the spatial-related questions in ExploraQA.

**QA Prompt:**

You are an intelligent question answering agent. I will ask you questions about an indoor space and you must provide an answer.

You will be shown a set of images that have been collected from a single location.

Given a user query, you must output `text` to answer to the question asked by the user.

User Query:

[ {img1}, {img2}, {img3} ] (Relative positions: img1 (x1, y1, z1, h1), img2 (x2, y2, z2, h2), img3 (x3, y3, z3, h3)) {question}

Figure 10: The prompt for QA VLM.

**Critic Prompt**

You are an AI assistant who will help me to evaluate the response given the question and the correct answer.

To mark a response, you should output a single integer between 1 and 5 (including 1, 5).

5 means that the response perfectly matches the answer.

1 means that the response is completely different from the answer.

Example 1: Question: Is it overcast?

Answer: no, Response: yes, Your mark: 1

Example 2: Question: Who is standing at the table?

Answer: woman, Response: Jessica, Your mark: 3

Example 3:

Question: Are there drapes to the right of the bed?

Answer: yes, Response: yes, Your mark: 5

**Your Turn:**

Question: {question}

Answer: {answer}

Response: {prediction}

Figure 11: The prompt for Critic VLM

