# OpenReview forum: "ExploraQA: Embodied Question Answering with Long-horizon Proactive Exploration"
_ICLR.cc/2026/Conference — ICLR 2026 Conference Withdrawn Submission_

### Official Review · Reviewer_BRMP · 2025-10-31

**Soundness:** 2
**Presentation:** 2
**Contribution:** 2
**Rating:** 4
**Confidence:** 3

**Summary:**

This paper proposes a new dataset and approach for embodied question answering in navigation settings. As a first contribution they curate a new QA dataset using an iterative data generation framework. The main difference between this dataset (and framework) is that it uses expert trajectory data to label some QA. This is what is being referred to as proactive QA. The whole premise being that expert data should be useful to train imitation learning policies. The data generation framework uses multiple VLMs in the loop where one model generates while the other verifies and several rounds happen before the answer is accepted. Using this data a planning framework (AQ Nav) is proposed. AQNav builds a topological map, a navigation module for next subgoal prediction. This waypoint (node) selection mechanism uses the textual information and the graph structure (which includes the visual features from all the different locations) to propose an action, which is either a node in the graph or stop action. The main sell of the approach is that the same expert data can be used for imitation learning as well as for asking QA questions.  Experiments are performed on the proposed ExploraQA dataset where the approach outperforms some old baslines and a more recent ETPONav baseline.

**Strengths:**

The paper is overall well written and tries to tackle an important problem. Curating high quality datasets for navigation is still challenging and its great to see a paper exploring it.

**Weaknesses:**

Questions:

In the related work, the paper mentions that the current paper “distinguishes itself by emphasizing the agent’s active question answering capability …. where the agent can proactively seek clarification or additional information”. But I am really sure how this proactive behavior shows up in any of the evals. Can you please clarify this? I think using the word “proactive” seems a bit misguided. Proactive question answering would assume that the agent has some self-awareness and can reason about uncertainty but that is not really what is being proposed here. From my understanding some of the proactive claims (e.g. proactive trajectories) should be substantially reduced.

*Need for a paired dataset*: One of the claims that the paper makes is the need for a paired dataset from which both navigation can be learned via imitation learning as well as have QA pairs to query for video-navigation QA. However, I am a bit unclear about this need. Why can we not learn both of these capabilities from disjoint datasets. Maybe some small amount of paired data is useful but it doesn’t seem super necessary to have a large scale dataset (which is what the paper is proposing) for these capabilities. Both, navigation and embodied QA capabilities seem disjoint. The agent can quite easily learn navigation from hindisght labeled data while embodied QA pairs can be generated from both optimal as well as suboptimal data. Can the authors clarify the need for this large scale paired dataset?

*Comparisons*: There are some recent works such as Saxena et al. which should be cited and compared against as well. Some of the techniques followed in these papers are quite similar to the proposed work

Saxena et al Grapheqa: Using 3d semantic scene graphs for real-time embodied question answering

**Questions:**

please see above

---

### Official Review · Reviewer_U8MS · 2025-11-01

**Soundness:** 1
**Presentation:** 2
**Contribution:** 1
**Rating:** 2
**Confidence:** 4

**Summary:**

The paper addresses the problem of asking an agent to follow some instructions to navigate to a target location, and then answer questions about the destination. It introduces a benchmark (ExploreQA) and an agent baseline (AQ-Nav).

To the best of my understanding, the ExploreQA task is as follows:
* Follow the instructions to navigate efficiently to a target location (using the instructions and trajectories from VLN-CE in the Habitat simualtor)
* Answer a question based on views near the end of the trajectory (generated using the proposed pipeline)

The authors also introduce the AQ-Nav agent and compare it to other baselines on the proposed benchmark

**Strengths:**

The details that are included in the paper are clearly presented, and the paper is well-organized. Some details (like prompts) are provided in the appendix.

The authors show VLM-as-a-judge evaluation has high agreement with human annotators, matching the findings in OpenEQA.

The AQ-Nav baselines has multiple components that the authors show an ablation of in Table 3.

**Weaknesses:**

My main concerns about the paper are
1. The problem introduced does not appear well-motivated to me, in that existing benchmarks already cover these capabilities
2. The benchmark adds some LLM-generated (and human checked) questions on top of existing trajectories from other works. Some details are missing about what design parameters were used to ensure the generation was complete (high-recall).
3. The proposed agent is evaluated only on this new benchmark; not on other existing benchmarks -- which makes it difficult to evaluate the efficacy of the agent.

**Task novelty and utility**
The proposed task asks the agent to follow a given trajectory, based on language instructions, and then answer a question based on what the agent sees at the destination. The episode is scored based on how efficiently the agent follows the trajectory, and whether the answer is correct.

Both the instruction-guided trajectory following [1] and question answering [EQA] tasks themselves are from existing work. Putting these two tasks together does not seem to require substantially different capabilities than what is already required in existing benchmarks:.
* The trajectories for instruction-guided trajectory are directly used from [1]
* The question-answering from a trajectory is also covered in OpenEQA: the EM-EQA setting requires agents to answer questions from a predetermined video, and the A-EQA setting requires agents to determine how to navigate around the environment to find the answer
Both of these benchmarks use the Habitat simulator.



**Question generation:**
There are very few details about the question generation provided.
* How do the authors guarantee coverage of useful or important questions that a user might ask? As best I can tell, the diversity comes from whatever GPT4o generates from the prompt in Figure 6.
* LLMs are known to collapse modes and produce relatively low diversity. For example, ask ChatGPT to tell a joke -- it will usually repeat the same 3 or 4 jokes.  So I wonder how the authors guarantee good coverage of what humans might ask in ExploreQA?

There are also few details about design decisions used to develop the QA evaluator:
* How many feedback iterations are does this usually take? How does acceptance rate increase with more iterations?
* What design decisions or facts about the prompt were important when designing the system?
* The authors state that this uses Claude Sonnet 3.5, which is known to have relatively weaker visual understanding compared to GPT-4, Qwen, or Gemini.


**Agent evaluation**
Does the agent generalize to other simulators and other question types, navigation instructions, or to longer trajectories?


For [1], it would also be helpful to compare the SotA on that benchmark, and ideally on other benchmarks in other simulators if they exist.


[1] Beyond the navgraph: Vision and language navigation in continuous environments

**Questions:**

* The authors mention "proactive navigation" and proactive several times but I am unfamiliar with the term. Is "proactive navigation" the task in VLN-CE?
* In S3.1, the paper says the viewpoints are selected s.t. "The relative orientation between poses is managed to guarantee non-overlapping visual coverage."

---

### Official Review · Reviewer_Pwv9 · 2025-11-01

**Soundness:** 2
**Presentation:** 3
**Contribution:** 2
**Rating:** 4
**Confidence:** 4

**Summary:**

This paper introduces ExploraQA, a new embodied question answering (EQA) benchmark designed to evaluate long-horizon, proactive, question-driven exploration in 3D embodied environments. The benchmark contains 12k question-answer pairs across seven categories, with longer navigation trajectories and multiple valid viewpoints per question. The authors also propose Answer Quality-Guided Navigation (AQ-Nav), which integrates a topological memory, keyframe selection mechanism, and LLM-based answer-quality reinforcement to improve question-guided exploration and answering. Experiments on Habitat environments show improvements in question-answering and navigation metrics over several baseline navigators.

**Strengths:**

- Interesting benchmark targeting long-horizon, question-conditioned exploration — relevant for embodied agents. Important since proactive exploration remains under-tested in embodied AI.

- Clear dataset effort: multiple viewpoints, more realistic navigation lengths, broader question categories.

- AQ-Nav idea (LLM-based answer-quality signal + keyframe filtering) is novel and conceptually interesting.

- Reasonable gains over baselines on their benchmark.

- Pipeline is well-engineered — topological memory + relevance filtering + RL reward shaping is a coherent architecture.

**Weaknesses:**

- LLM generation vs human verification tradeoff is unclear. If every QA pair still goes through humans, where is the true scalability advantage? ~17% rejection rate suggests humans are still bottleneck. LLMs may also reduce linguistic diversity (mode collapse), undermining benchmark diversity. Why not human-written questions if humans are reviewing anyway?

- The introduction (correctly) points out that other benchmarks like OpenEQA have “passive trajectories” that are ineffective as expert demonstrations for imitation learning, motivating the need for “proactive trajectories.” However, the paper does not include any imitation-learning evaluation to support this claim, making the stated IL motivation currently unverified.

- No real analysis of diversity or difficulty of QA pairs. Benchmark value depends on distributional richness and question complexity. No evaluation that LLM-generated QA isn't repetitive or too easy.

- Experiments: No comparison to simple baselines like a blind LLM or a multi-frame VLM with randomly sampled frames from the scene.

- Human verification details limited; no quality statistics beyond a correlation metric. No inter-annotator agreement scores

- No cost analysis (GPU hours, human annotation effort, CO2 footprint)

**Questions:**

1. LLM-generated QA vs human QA: As discusses above, If all QA pairs require human verification anyway, what is the actual impact/efficiency gain? Did you measure linguistic diversity / question quality vs human-written questions?

2.  Question collapse risk
- Did you analyze entropy / lexical diversity / syntactic variety of generated questions?
- Is there evidence your dataset avoids LLM-mode-collapse phrasing patterns?

3. Imitation learning motivation
- You argue passive trajectories aren't suitable for imitation learning (and proactive exploration helps) — do you actually train an IL agent on ExploraQA to validate this? If not, can you soften the claim or add discuss IL results?

4. Reward shaping
- Why reward only score==5?
- Did lower thresholds or soft rewards fail?

5. Baselines
- I'd really recommend adding simpler baselines, in particular (1) blind LLMs (2) Multi-frame VLMs and (3) Human baselines to give estimates of lower and upper bound of the benchmark and some sense of the difficulty. This is standard practice, as done in previous EQA benchmarks like OpenEQA.

6. Human workload
- What is the total hours / cost for human verification? Is the ~17% correction rate stable across scenes?

---

### Note · Authors · 2026-01-28

I have read and agree with the venue's withdrawal policy on behalf of myself and my co-authors.

---

### Meta-Review · Area_Chair_BDW9 · 2025-12-26

**Summary:**

The reviewers expressed a range of concerns regarding the claims, experimental evaluation, baseline comparisons, novelty, and computational complexity. The authors did not submit a rebuttal and therefore did not engage with any of these questions. As a result, none of the reviewers’ concerns were addressed, clarified, or resolved, leaving significant questions about the validity and contribution of the work unanswered.

**Reviewer Concerns:**

The authors did not submit a rebuttal and therefore did not engage with any of these questions. As a result, none of the reviewers’ concerns were addressed, clarified, or resolved, leaving significant questions about the validity and contribution of the work unanswered.

**Reviewer Scores:**

The reviewers’ scores are expected to remain on the negative side and recommend rejection.

---

### Decision · Program_Chairs · 2026-01-26

Reject